# Path Tracing-Inspired Modeling of Non-Line-of-Sight SPAD Data

**DOI:** 10.3390/s24206522

**Published:** 2024-10-10

**Authors:** Stirling Scholes, Jonathan Leach

**Affiliations:** School of Engineering and Physical Sciences, Heriot-Watt University, Edinburgh EH14 4AS, UK

**Keywords:** non-line of sight, imaging, single-photon avalanche diode, NLOS, SPAD, Lidar

## Abstract

Non-Line of Sight (NLOS) imaging has gained attention for its ability to detect and reconstruct objects beyond the direct line of sight, using scattered light, with applications in surveillance and autonomous navigation. This paper presents a versatile framework for modeling the temporal distribution of photon detections in direct Time of Flight (dToF) Lidar NLOS systems. Our approach accurately accounts for key factors such as material reflectivity, object distance, and occlusion by utilizing a proof-of-principle simulation realized with the Unreal Engine. By generating likelihood distributions for photon detections over time, we propose a mechanism for the simulation of NLOS imaging data, facilitating the optimization of NLOS systems and the development of novel reconstruction algorithms. The framework allows for the analysis of individual components of photon return distributions, yielding results consistent with prior experimental data and providing insights into the effects of extended surfaces and multi-path scattering. We introduce an optimized secondary scattering approach that captures critical multi-path information with reduced computational cost. This work provides a robust tool for the design and improvement of dToF SPAD Lidar-based NLOS imaging systems.

## 1. Introduction

Since the early demonstrations of ‘Seeing Around Corners’ [1], Non-Line of Sight (NLOS) imaging has become an active area of research; see Refs [2,3] for recent reviews. Specifically, the ability to detect and/or track and reconstruct objects beyond the line of sight of an imaging system, based on the light scattered by the objects, has applications ranging from surveillance to autonomous navigation. NLOS is commonly realized using systems comprised of two principle parts: an active imaging system, in the form of a Lidar, and a reconstruction algorithm.

The first part, the active imaging system, is responsible for measuring information about the scene. This is often performed using a direct Time of Flight (dToF) Lidar system, in which pulses of light are used to illuminate the scene whilst a synchronized detector captures the scattered light. By accumulating the arrival time of the signal photons into a histogram, the total path length of the measured photons is constrained, enabling the 3D shape of the scene to be reconstructed. Single-Photon Avalanche Diodes (SPADs), when combined with Time-to-Digital Converters (TDCs) are well-suited to this task for two reasons: first, their single-photon sensitivity allows for the detection of signal photons despite the losses due to multiple scattering events [4,5,6]; second, their ability to time-tag photon arrivals enables the inverse reconstruction algorithms required by NLOS [7,8]. Consequently, SPAD Lidar-based NLOS has been realized in a variety of regimes, such as the following: using SPAD-array cameras operating in Short-Wave InfraRed (SWIR) [9,10]; with novel SPAD triggering regimes, to reduce range ambiguity [11]; improving localization via temporal focusing [12]; using compact commercially available SPAD sensors [13,14]; and at ranges from 50 m to 1.4 km [5,15]. Whilst dToF SPAD Lidars are the most common form of NLOS imaging systems, other Lidar configurations have been demonstrated, such as Frequency-Modulated Continuous Wave (FMCW) systems based on optical combs [16], THz radiation-based systems [17], Super-conducting Nano-wire Single-Photon Detector (SNSPD)-based systems [18], non-linear wavelength conversion systems [19,20], and structured light-based systems [21].

The second part in an NLOS system is the reconstruction algorithm, which works in conjunction with the activate imaging system. The algorithm is responsible for processing the time-of-arrival information from the imaging system in such a way that the 3D position of objects can be reconstructed. A number of NLOS image-reconstruction algorithms have been presented; for instance: light cone transformations when using confocal systems [22,23]; approaches based on the first measured photon [24]; improved occlusion and jitter resistance [25,26]; reduced photon accumulation and sample point requirements [27,28]; and the reconstruction of color images [29]. Furthermore, approaches aimed at realizing real-time processing using phasor fields, low-latency algorithms, and forward projection optimization have been presented [30,31,32], with recent processing techniques including the use of machine learning algorithms [33,34].

Although NLOS has been realized in several publications, relatively little formalism has been presented to predict the performance of SPAD-based NLOS systems [15,35,36]. Prior works on simulating NLOS systems have explored various approaches, including the use of simulated 3D environments [37], simulations of scanning systems [38,39], and simulations of light cone returns [40]. However, simulating the temporal distribution of photon detections at a sensor for scenarios featuring multiple objects in room-like environments remains challenging. Furthermore, it has been shown that once the temporal distribution of photon detections is known, SPAD data can be accurately simulated [41]. Data of this type can then be used to refine the components of imaging systems, develop novel reconstruction algorithms, and create machine learning data sets for training neural networks.

In this work, we present a versatile framework for modeling the temporal distribution of photon detections in dToF Lidar systems in the context of NLOS imaging. We demonstrate a proof-of-principle simulation via the creation of a virtual environment within the Unreal Engine. The simulation is built in two steps, as shown in Figure 1a. Firstly, the attenuation and path length from the laser spot to the SPAD detector is determined for a large number of sample points in the room. Secondly, by convolving the IRF of the detector (assumed in this work to be governed by a laser pulse with a Gaussian distribution in time) with the path length and attenuation the temporal distribution of photon detections at the SPAD is simulated. Our path tracing-inspired approach is able to account for a number of important factors, including the material reflectivities of objects, their distance from sources of illumination, and object occlusion. When combined, these quantities provide likelihood distributions for photon detections as a function of time, enabling the simulation of SPAD detection likelihoods. Additionally, the use of the computational model allows the individual components of the photon return to be isolated and the impact of extended features, i.e., walls, ceilings, and floors, to be commented on. Furthermore, we examine the impact of multi-pathing, in terms of the distribution of photon detections and on computational complexity. We address the latter of these challenges by implementing an optimized secondary scattering approach, which captures the salient multi-pathing information at a reduced computational cost.

## 2. Problem Description

This work makes use of a ‘demonstration room’, as shown in Figure 1, featuring a simple single ‘L’ bend at the end of a corridor, to function as a corner. The room contains two objects. Firstly, a vertical ‘pillar’, colored orange in Figure 1, which runs from the floor to the ceiling. This pillar has been rotated, such that its surfaces are at $45^{\circ }$ with respect to the walls of the room. This rotation is to examine the effect of non-orthogonal surfaces on the scattering. Secondly, the room contains a ‘box’, colored purple in Figure 1. The box primarily serves to cast a shadow into the room, to examine the effect of object occlusion on the system. Although we have used this simple room for a proof-of-principle demonstration, the implementation of our approach within a game development environment means that the large catalog of readily available 3D environments developed for games could be directly leveraged to explore NLOS in a diversity of scenarios that would be impractical to realize experimentally.

The SPAD-based dToF Lidar system is assumed to be co-located at a point in the corridor leading to the room shown in Figure 1a. The pulse laser illuminates a single point, referred to as the ‘laser hit point’ (the red dot in Figure 1a) [42,43]. Figure 1 also shows the implementation of the room within the Unreal game engine. In Figure 1b, the laser point is represented by the light bulb icon. The shadows visible in the Unreal environment are cast by this light source. Unfortunately, the lighting within the virtual environment is not sufficiently physical to be matched directly with real systems. The SPAD is assumed to be coupled to a lens system that images an area, i.e., a Field of View (FoV), as shown by the green square in Figure 1. This modeling scheme is compatible with both pixel array-type sensors [44,45,46,47,48] and single-pixel sensors, such as SNSPDs [49]. Note that throughout this work a left-handed coordinate system is used. This is to maintain consistency with the Unreal engine, which also uses a left-handed coordinate system.

## 3. Simulation Framework

### 3.1. Sampling the Scene

The scattering of light in an environment can be approximated by dividing the scene into a large number of points and determining the relationship between these points and the light source. This section outlines a framework for decomposing a 3D environment into a scalable number of points, as well as determining the properties, i.e, material, distance *R*, and angle to the light source. In conjunction with Figure 2a, consider a laser pulse *I* incident at a laser hit point ($x_{1},y_{1},z_{1}$). The scattering of the pulse off of the laser hit point ($x_{1},y_{1},z_{1}$), shown as red lines in Figure 2a, can be characterized by a hemispherical expansion ($1/R^{2}$), a reflectivity coefficient ($\Gamma_{1}$), and an angular distribution function referred to as a Bi-directional Reflectance Distribution Function (BRDF) ($B_{1}$) [50]. For a point ($x_{j},y_{j},z_{j}$) in the room, at a distance $R_{j}$ from the laser hit point, the amplitude $A_{j}$ of the pulse is
(1)$$\begin{array}{c} A_{j}=I\times \frac{\Gamma_{1}\alpha_{1}B_{1}\left(\theta_{1}^{i},\phi_{1}^{i},\theta_{1}^{r},\phi_{1}^{r}\right)}{R_{j}^{2}}. \end{array}$$

Here, $\theta_{1}^{i},\phi_{1}^{i}$ denote singular incident angles, and $\theta_{1}^{r},\phi_{1}^{r}$ denote singular reflected angles for $B_{1}$ associated with the laser hit point. Figure 2b illustrates these angles from the perspective of $B_{1}$; $\alpha_{1}$ is a normalization constant, to ensure the conservation of energy. The θ angles span a 2π range in the local $y_{1}$, $z_{1}$ plane. The ϕ angles span the range [0,π/2] defined relative to the surface normal $x_{1}$. The BRDF takes in these angles as arguments and returns what fraction of the energy incident at a point from a specific direction $\theta^{i},\phi^{i}$ is scattered in a specific reflection direction $\theta^{r},\phi^{r}$.

To approximate the hemispherical scattering of the laser hit point, a large number *N* of rays $R_{j}|j\in \left[1,N\right]$ are projected into the room from the laser hit point. These rays are realized within the Unreal Engine, using line traces. The ray length $R_{j}$ is defined as the Euclidean distance between the laser hit point ($x_{1},y_{1},z_{1}$) and the intersection point of the ray with the environment ($x_{j},y_{j},z_{j}$):(2)$$\begin{array}{c} R_{j}=\sqrt{{\left(x_{1}-x_{j}\right)}^{2}+{\left(y_{1}-y_{j}\right)}^{2}+{\left(z_{1}-z_{j}\right)}^{2}}. \end{array}$$

At each point ($x_{j},y_{j},z_{j}$) where the ray $R_{j}$ intersects the environment a ‘lookin’ is created. A lookin is a construct that ‘looks into’ the room, and it is realized in the Unreal Engine by spawning a camera at the point ($x_{j},y_{j},z_{j}$) aligned with the surface normal. This camera has a 180° FoV (to simulate hemispherical scattering) and a transverse resolution of $\tilde{N}=l\times p$ total pixels. The camera performs multiple render passes to extract the surface normal, the surface material, and the distance to the nearest surface for each pixel. Each pixel is then considered a new sample of the room, referred to as an ‘observed point’, as shown by the green lines in Figure 2a, creating a new set of $\tilde{N}$ observed points ($x_{k}^{o},y_{k}^{o},z_{k}^{o}|k\in \left[1,\tilde{N}\right]$). This combination of line traces and camera spawning allows a total of $N\times \tilde{N}$ observation points of the room to be gathered more efficiently than using an equivalent number of line traces. Each lookin is instantiated with a reflectivity $\Gamma_{j}$ and a BRDF $B_{j}$ based on the material properties and the surface normal of the point ($x_{j},y_{j},z_{j}$). For each $B_{j}$, a corresponding set of local angles ($\theta_{j}^{i},\phi_{j}^{i},{\hat{\theta }}_{j}^{r},{\hat{\phi }}_{j}^{r}$) is created, where ${\hat{\theta }}_{j}^{r},{\hat{\phi }}_{j}^{r}$ denote the reflected angles to the set $\tilde{N}$ of all observed points. Throughout this work, the ^ notation is used to indicate a collection of values, such as all the angles to the collection of observed points from a given lookin. To calculate the energy sent from the laser hit point to each observed point ($x_{k}^{o},y_{k}^{o},z_{k}^{o}$), the observed points must be converted from the coordinate system of the lookin point to the coordinate system of the laser hit point ($x_{k}^{lsr},y_{k}^{lsr},z_{k}^{lsr}$). This is achieved by applying a 3-dimensional coordinate transformation matrix $\Omega_{lsr}$ to the points: (3)$$\begin{array}{c} \left(\begin{array}{c} x_{k}^{lsr}\\ y_{k}^{lsr}\\ z_{k}^{lsr} \end{array}\right)=\Omega_{lsr}*\left(\begin{array}{c} x_{k}^{o}\\ y_{k}^{o}\\ z_{k}^{o} \end{array}\right)-\Omega_{lsr}*\left(\begin{array}{c} x_{j}\\ y_{j}\\ z_{j} \end{array}\right). \end{array}$$

The transformation $\Omega_{lsr}$ must be calculated and applied independently for each lookin, as it has an implicit dependence on the surface normal of the lookin point to which it is associated. By combining Equations (Equation 2) and (Equation 3), ${\hat{\phi }}_{1}^{r}$ can be derived from the dot product as
(4)$$\begin{array}{c} {\hat{\phi }}_{j}^{r}=\arccos\left\{\frac{-[{\hat{n}}_{x}^{lsr}\left(x_{k}^{lsr}\right)+{\hat{n}}_{y}^{lsr}\left(y_{k}^{lsr}\right)+{\hat{n}}_{z}^{lsr}\left(z_{k}^{lsr}\right)]}{\sqrt{{\left(x_{k}^{lsr}\right)}^{2}+{\left(y_{k}^{lsr}\right)}^{2}+{\left(z_{k}^{lsr}\right)}^{2}}}\right\}|k\in \left[1,\tilde{N}\right], \end{array}$$
where ${\hat{n}}^{lsr}$ is the normal vector associated with the laser hit point. The negative sign is present to account for the reflection; $\theta_{j}^{r}$ is given by
(5)$$\begin{array}{c} {\hat{\theta }}_{j}^{r}=\arctan4\left\{y_{k}^{lsr},z_{k}^{lsr}\right\}|k\in \left[1,\tilde{N}\right], \end{array}$$
where arctan4 represents the four-quadrant arctan function.

Figure 3 illustrates the distribution of the laser energy throughout the room to all observed points $N\times \tilde{N}$. For illustrative purposes, the room is sampled using a 10×10 grid of rays $R_{j}$, resulting in N=100 lookin points, with each lookin point sampling the room using 60×60 pixels for $\tilde{N}=3600$, resulting in a total of 360,000 observation points of the room. The left panel shows the inverse-square loss as the distance from the laser hit point increases. The center panel shows the energy distribution based on the BRDF associated with the laser hit point, which is assumed to be Lambertian, i.e., $B_{1}\left(\theta_{1}^{i},\phi_{1}^{i},{\hat{\theta }}_{1}^{r},{\hat{\phi }}_{1}^{r}\right)=\cos\left({\hat{\phi }}_{1}^{r}\right)$. The right panel shows the combined effects of the inverse-square loss and the BRDF. For Lambertian scattering, the arccos function from Equation (Equation 4) is effectively removed.

### 3.2. Primary Scattering Paths

Figure 4 illustrates the implementation of the primary scattering paths. For each lookin point ($x_{j},y_{j},z_{j}$) the local incident angles ($\theta_{j}^{i},\phi_{j}^{i}$) from the laser point are calculated together with the reflection angles (${\hat{\theta }}_{j}^{r},{\hat{\phi }}_{j}^{r}$) to each observed point ($x_{k}^{o},y_{k}^{o},z_{k}^{o}|k\in \left[1,\tilde{N}\right]$). The BRDF, abbreviated as “*B*”, for each lookin $B_{j}$ is calculated, using the local angles. Some fraction of the observed points for each lookin fall within the FoV of the SPAD. The energy from this reduced set of observed points is scattered back to the sensor. This ‘final’ scattering is proportional to the BRDFs of the observed points within the FoV ($\left[B_{k}^{o}:B_{n}^{o}\right]$) and the paths back to the sensor $\left[R_{k}^{s}:R_{n}^{s}\right]$.

Additionally, while a lookin point cannot be spawned in a shadowed region of the room, it is possible for an observed point to lie within a shadowed region, as illustrated by the red cross in the shadowed region of Figure 4. To address this, observed points that lie in shadow must be removed from calculations involving the energy distributed from the initial laser hit point ($x_{1},y_{1},z_{1}$). The calculation of the final scattering is mechanically similar to the calculations for prior scattering points. A rotation matrix $\Omega_{k}^{o}$ is used to transform the location of the lookin point ($x_{j},y_{j},z_{j}$) to the coordinate system of the observed point, such that the incident angles $\left[\theta_{k}^{i}:\theta_{n}^{i}\right]$ and $\left[\phi_{k}^{i}:\phi_{n}^{i}\right]$ can be calculated. Additionally, the reflection angles $\left[\theta_{k}^{r}:\theta_{n}^{r}\right]$ and $\left[\phi_{k}^{r}:\phi_{n}^{r}\right]$ from each observed point within the FoV back to the position of the sensor are calculated. These angles are passed to the BRDF for each observed point within the SPAD FoV, which, together with the reflectivity coefficients Γ, defines the energy scattered back to the sensor.

Figure 5 shows the steps in the primary scattering scheme for two lookin points shown as gray triangles. Figure 5a shows the energy distribution for a lookin point located on the nearest wall with a Lambertian BRDF, such that $B_{j}=\cos\left({\hat{\phi }}_{j}^{r}\right)$. Figure 5b shows the energy distribution for a lookin point located on the pillar face nearest the laser hit point. For illustrative purposes, the pillar has been assigned a BRDF of $B_{j}=\sin\left({\hat{\theta }}_{j}^{r}\right)$, which causes it to direct the majority of its energy into a plane parallel with the floor. Furthermore, the observed points from the pillar lookin have a ‘triangular’ shape as a result of the $180^{\circ }$ FoV of the lookin combined with the angled surface upon which it has been spawned. The green square illustrates the FoV of the SPAD. The observation points that fall within the green square are used to calculate the final bounce back to the SPAD. The left panel shows the BRDF energy distributions from their respective lookin points. In comparison to the center panel of Figure 3, the left column of Figure 5a illustrates the same cosine distribution, although it has now been rotated to align with the surface normal of the lookin point. The center panel shows the effect of the inverse square scaling on the energy distributions. The right panel shows close-in views of the SPAD FoV. These panels also illustrate the relative scattering intensity of each observed point, including the final bounce calculation. Once the final bounce has been calculated, the likelihood envelope function $L_{p}\left(t\right)$ in time *t* associated with the primary paths can be written as the convolution of the path length and attenuation with an IRF, as follows:(6)$$\begin{array}{cc} \; & A_{1}=\frac{\Gamma_{1}C_{atm}^{R_{j}}\alpha_{1}B_{1}\left(\theta_{1}^{i},\phi_{1}^{i},\theta_{1}^{r},\phi_{1}^{r}\right)}{{\left(R_{j}\right)}^{2}},\\ \; & A_{2}=\frac{\Gamma_{j}C_{atm}^{R_{k}^{o}}\alpha_{j}B_{j}\left(\theta_{j}^{i},\phi_{j}^{i},\theta_{j}^{r},\phi_{j}^{r}\right)}{{\left(R_{k}^{o}\right)}^{2}},\\ \; & A_{3}=\frac{\Gamma_{k}^{o}C_{atm}^{2R_{k}^{s}}\beta_{k}^{o}B_{k}^{o}\left(\theta_{j}^{i},\phi_{j}^{i},\theta_{j}^{r},\phi_{j}^{r}\right)}{{\left(R_{k}^{s}\right)}^{2}},\\ \; & L_{p}\left(t\right)=q\pi {\left(\frac{f}{2f_{no}}\right)}^{2}\sum_{\left[k:n\right]}I\times A_{1}\times A_{2}\times A_{3}\times \frac{1}{\sigma \sqrt{2\pi }}\exp\left\{-\frac{1}{2}{\left[\frac{t-(R_{j}+R_{k}^{o}+R_{k}^{s})}{\sigma }\right]}^{2}\right\}, \end{array}$$
where σ represents the width of the impulse response function of the SPAD to a laser pulse with a Gaussian temporal shape, *q* is the quantum efficiency of the SPAD detector, and $C_{atm}$ is the atmospheric attenuation term within the room. The β term is a modified normalization that accounts for the single input and output vectors of the bounces associated with ${\hat{A}}_{3}$; *f* is the focal length of the collecting lens; $f_{no}$ is the f-number of the collection lens; *f* and $f_{no}$ are used together, to define the aperture of the collection optics. This aperture term is present in Equation (Equation 6), to render $L_{p}\left(t\right)$ unitless.

## 4. Secondary Scattering Paths

To extend the modeling to multiple bounces, a criteria that all secondary paths must pass through the associated lookin point is enforced, as illustrated in Figure 6. For a given lookin point ($x_{j},y_{j},z_{j}$), the local incident angles are expanded to the set (${\hat{\theta }}_{j}^{i},{\hat{\phi }}_{j}^{i}$) to accommodate the illuminated (i.e., not in shadow) observed points ($x_{k^{\prime }}^{o},y_{k^{\prime }}^{o},z_{k^{\prime }}^{o}|k^{\prime }\in \left[1,\tilde{N}\right]$). Due to the reciprocal nature of the BRDF function, the incident angles (${\hat{\theta }}_{j}^{i},{\hat{\phi }}_{j}^{i}$) from the observed points to the lookin point are the equivalent of the reflection angles (${\hat{\theta }}_{j}^{r},{\hat{\phi }}_{j}^{r}$) from the lookin point to the set of observed points. Consequently, a coordinate system transform Ω is not required. Each of the observed points contributes a scattering component proportional to the BRDFs of the observed points ($B_{k^{\prime }}^{o}|k^{\prime }\in \left[1,\tilde{N}\right]$), their coefficients of reflection ($\Gamma_{k^{\prime }}^{o}|k^{\prime }\in \left[1,\tilde{N}\right]$), and the total path length ($R_{k^{\prime }}^{o}+R_{k^{\prime }j}^{o}|k^{\prime }\in \left[1,\tilde{N}\right]$) from the laser hit point to the lookin point.

Characterizing secondary paths using the approach shown in Figure 6 has three effects. Firstly, the longer path length traveled ($R_{k^{\prime }}^{o}+R_{k^{\prime }j}^{o}$) ensures that the contributions from secondary paths are always less than a primary path, even for cases where $B_{k^{\prime }}^{o}$ is a perfect specular reflector. Secondly, the longer path length also ensures that the contributions from secondary paths arrive at the detector later in time than the primary paths associated with each lookin. Thirdly, the computational complexity of the problem now scales as $O({\tilde{N}}^{2})$. For instance, the set of total path lengths ${\hat{T}}_{j}$ for a single lookin point from the laser hit point to the observed points within the FoV of the SPAD is given in vector form by
(7)$$\begin{array}{cc} {\hat{T}}_{j} & =\left(\begin{array}{ccccc} T_{1,1} & . & . & . & T_{1,n^{\prime }}\\ . & . & \; & & .\\ . & \; & . & \; & .\\ . & \; & & . & .\\ T_{n,1} & . & . & . & T_{n,n^{\prime }} \end{array}\right)\\ \;\\ \; & =\left(\begin{array}{c} R_{k}^{o}\\ .\\ .\\ .\\ R_{n}^{o} \end{array}\right)⨁{\left(\begin{array}{c} R_{k^{\prime }}^{o}+R_{k^{\prime }j}^{o}\\ .\\ .\\ .\\ R_{n^{\prime }}^{o}+R_{n^{\prime }j}^{o} \end{array}\right)}^{T}\\ \;\\ \; & =\left(\begin{array}{ccccc} R_{k}^{o}+R_{k^{\prime }}^{o}+R_{k^{\prime }j}^{o} & . & . & . & R_{k}^{o}+R_{n^{\prime }}^{o}+R_{n^{\prime }j}^{o}\\ . & . & \; & & .\\ . & \; & . & \; & .\\ . & \; & & . & .\\ R_{n}^{o}+R_{k^{\prime }}^{o}+R_{k^{\prime }j}^{o} & . & . & . & R_{n}^{o}+R_{n^{\prime }}^{o}+R_{n^{\prime }j}^{o} \end{array}\right).\; \end{array}$$

In Equation (Equation 7), the *T* represents the transpose and the ⨁ represents the vector addition operation. Equation (Equation 7) illustrates that for each path from the lookin point to an observed point within the FoV of the SPAD, all contributions from all observed points must be considered, with analogous operations for Equation (Equation 7) existing for the BRDFs. This $O({\tilde{N}}^{2})$ dependence results in poor computational scalability, even at relatively limited sampling densities ($N,\tilde{N}$). The complete likelihood envelope function L(t) in time *t* can be written as the sum of the primary envelope function $L_{p}\left(t\right)$ and a secondary envelope function $L_{s}\left(t\right)$,
(8)$$\begin{array}{cc} \; & {\hat{A}}_{1}=\frac{\Gamma_{1}C_{atm}^{R_{k^{\prime }}^{o}}\alpha_{1}B_{1}\left(\theta_{1}^{i},\phi_{1}^{i},{\hat{\theta }}_{1}^{r},{\hat{\phi }}_{1}^{r}\right)}{{\left(R_{k^{\prime }}^{o}\right)}^{2}}|k^{\prime }\in \left[1,\tilde{N}\right],\\ \; & {\hat{A}}_{2}=\frac{\Gamma_{k^{\prime }}C_{atm}^{R_{k^{\prime }j}^{o}}\beta_{k^{\prime }}B_{k^{\prime }}\left(\theta_{k^{\prime }}^{i},\phi_{k^{\prime }}^{i},\theta_{k^{\prime }}^{r},\phi_{k^{\prime }}^{r}\right)}{{\left(R_{k^{\prime }j}^{o}\right)}^{2}}|k^{\prime }\in \left[1,\tilde{N}\right],\\ \; & {\hat{A}}_{3}=\frac{\Gamma_{j}C_{atm}^{R_{k}^{o}}\alpha_{j}B_{j}\left({\hat{\theta }}_{j}^{i},{\hat{\phi }}_{j}^{i},\theta_{j}^{r},\phi_{j}^{r}\right)}{{\left(R_{k}^{o}\right)}^{2}},\\ \; & A_{4}=\frac{\Gamma_{k}^{o}C_{atm}^{2R_{k}^{s}}\beta_{k}^{o}B_{k}^{o}\left(\theta_{j}^{i},\phi_{j}^{i},\theta_{j}^{r},\phi_{j}^{r}\right)}{{\left(R_{k}^{s}\right)}^{2}},\\ \; & L_{s}\left(t\right)=q\pi {\left(\frac{f}{2f_{no}}\right)}^{2}\sum_{\left[k:n\right]}\sum_{\left[k^{\prime }:n^{\prime }\right]}I\times \hat{A_{1}}\times \hat{A_{2}}\times \hat{A_{3}}\times A_{4}\times \frac{1}{\sigma \sqrt{2\pi }}\exp\left\{-\frac{1}{2}{\left[\frac{t-{\hat{T}}_{j}}{\sigma }\right]}^{2}\right\},\\ \; & L\left(t\right)=L_{p}\left(t\right)+L_{s}\left(t\right). \end{array}$$

Equation (Equation 8) is an extended form of Equation (Equation 6), to accommodate the additional bounce and $O({\tilde{N}}^{2})$ nature of the secondary paths.

Figure 7 shows the likelihood envelope function L(t) plotted on a log scale for a single laser hit point. This distribution also represents the histogram distribution that would be obtained by a SPAD sensor in the limit of infinite exposure time. To explore the impact of sampling density, the simulation was performed at two different resolutions. First, using a 10 × 10 grid of rays $R_{j}$, with each of the 100 lookin points sampling the room on a 60 × 60 grid. Second, using a 45 × 45 grid of rays $R_{j}$, with each of the 2025 lookin points again using a 60 × 60 sampling grid. For illustrative purposes, the effective aperture of the lens system in Equation (Equation 8) is assumed to have a radius of 20 mm. Additionally, the values of *q*, Γ, and $C_{atm}$ are all assumed to be one with σ=250 ps. The likelihood envelope function L(t) is the uppermost solid black line. The color-coded lines illustrate the contributions to the total envelope of each feature of the room—specifically, the walls, roof, and floor (blue), the box (purple), the pillar (orange), and the secondary bounce (red). Examining the purple and orange components of Figure 7, it can be seen that the signal returned from the pillar generally arrives both with a lower amplitude and a greater delay than the signal returned from the box, which is consistent with the pillar being further from the laser hit point than the box. Furthermore, these observations are consistent with the prior experimental works presented in Refs [5,6,51,52]. This consistency, together with the known feasibility of the likelihood approach to SPAD modeling [53], suggests that the approach presented here will agree with and could be used to model future SPAD-based NLOS experiments. However, complete validation of the proposed model will require quantitative comparisons to experiments featuring cluttered and dynamic environments, something beyond the immediate scope of this work. Additionally, examining only the pillar and box components of the distribution is consistent with prior experiments in which extended surfaces, i.e., the walls, floor, and roof of the room, are negligible. These extended surfaces are negligible in cases where either the materials of the extended surfaces are much less reflective than the objects in the room [18,25] or the distance between the objects in the room and the extended surfaces is large, relative to the distance between the object and the laser hit point [5,26]. When these conditions are not met, the total return is dominated by the reflections from extended surfaces, as shown by the agreement between the dashed blue line of the room reflections and the solid black line of the total envelope. The red line in Figure 7 confirms that the secondary paths make only a minor contribution to the total envelope, which is consistent with the energy loss due to additional scattering events. Figure 7 also shows that while denser sampling of the room smooths and somewhat broadens the envelope functions the overall profile remains relatively consistent. Finally, detection distributions of the type shown in Figure 7 are compatible with existing SPAD Lidar simulation techniques for generating physically realistic data [41].

### Optimized Secondary Scattering Paths

Figure 8 illustrates the implementation of the optimized secondary paths scheme. The scheme is identical to that described in Section 4; however, by only considering a subset of secondary paths, the computational complexity can be significantly reduced. This creates a scalable approach, in which the number of secondary paths considered for each lookin point can be adjusted based on the computational resources available and the density of room sampling required. Here, we demonstrate a single secondary path, i.e., once the secondary paths have been calculated, only the maximum secondary path is retained for the remaining calculations. This scheme significantly reduces the computational complexity of the simulation by returning the model to an O(N) problem. Specifically, for each path from the lookin point to an observed point within the FoV of the SPAD, only a single contribution from all observed points is considered, i.e., the transposed vector from Equation (Equation 7) is reduced from a vector to the singular value $R_{k^{\prime }}^{o^{\prime }}+R_{k^{\prime }j}^{o^{\prime }}$.

Additionally, for BRDFs that are independent of the incoming angles of the incident radiation, such as Lambertian scattering, the optimized path scheme has the effect of transforming the secondary envelope function $L_{s}\left(t\right)$ into a time-delayed and lower-amplitude copy of the primary envelope function $L_{p}\left(t\right)$. This is because the scattering from the lookin point is agnostic to the source of the incident radiation, i.e., a single incident ray from the laser hit point and a single incident ray from a secondary path are scattered through the same set of reflection angles. The additional bounce and the path length associated with the secondary path, therefore, adjust only the amplitude and the arrival time of the secondary path component but not the overall shape of the envelope.

Figure 9 shows the likelihood envelope function L(t) in the optimized secondary scattering scheme. For comparability with Figure 7, the same plotting conventions have been used. Figure 9 confirms that the optimized secondary path scheme retains features of the secondary path envelope $L_{s}\left(t\right)$ without effecting the primary envelope. Furthermore, since the total number of secondary paths being considered is reduced, the maximum amplitude of the secondary path envelope $L_{s}\left(t\right)$ is reduced, as is its total width. However, the position of the peak of $L_{s}\left(t\right)$ is unchanged, since the optimized scheme retains the most significant secondary scattering events. The precise increase in computation speed when using the optimized scattering approach depends on the total number of sample points being used. For the 10×10×60×60 case, a reduction in computation time from ≈ 1 hour to <30 s is observed. This increased processing speed allows for either the testing of more environments in a given time or the same environment to be sampled more densely in the same time.

## 5. Conclusions

This work presents a versatile framework for modeling the temporal distribution of photon detections in a dToF Lidar system. We have demonstrated a proof-of-principle simulation by means of a demonstration room within the Unreal Engine. Our approach is able to account for a number of important factors in the scene, and it provides likelihood distributions for photon detections as a function of time, thus enabling the further optimizing of NLOS imaging systems and the development of novel reconstruction algorithms. By analyzing individual components of this distribution, we explore the impact of extended surfaces. We examine the effects of multi-pathing and sampling density, demonstrating an optimized secondary scattering approach that captures the salient multi-pathing information at a reduced computational cost. We expect this work to be a useful addition to the dToF Lidar community by assisting in the design of NLOS imaging systems.

## Figures and Tables

**Figure 1 sensors-24-06522-f001:**
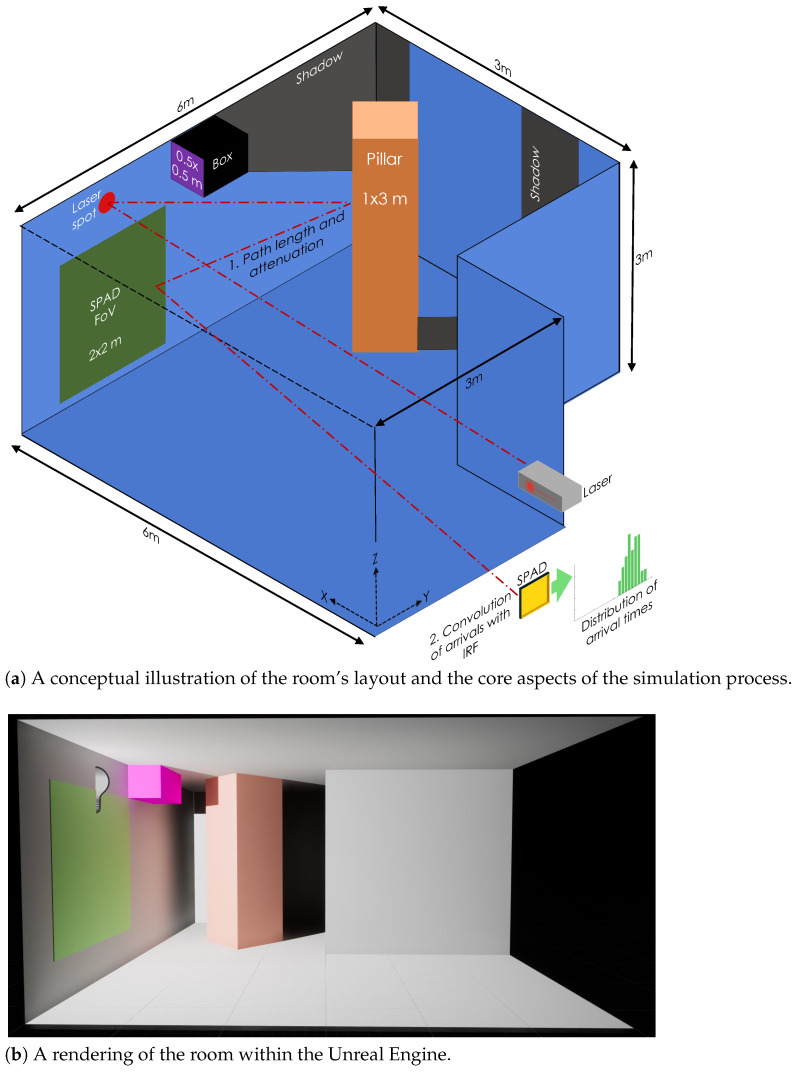
The demonstration room used throughout this work: (**a**) The components of the room together with their sizes. The simulation relies on determining the attenuation and path length from the laser spot to the SPAD detector. This information is then combined with an Impulse Response Function (IRF) to create temporal distributions of photon detections. (**b**) The room within the Unreal Engine. Throughout this work, the nearest corridor wall has been removed in graphics, to aid in visualization, but is accounted for in all calculations. Note the use of a left-hand coordinate system to ensure compatibility between the Unreal environment and the later data-processing steps.

**Figure 2 sensors-24-06522-f002:**
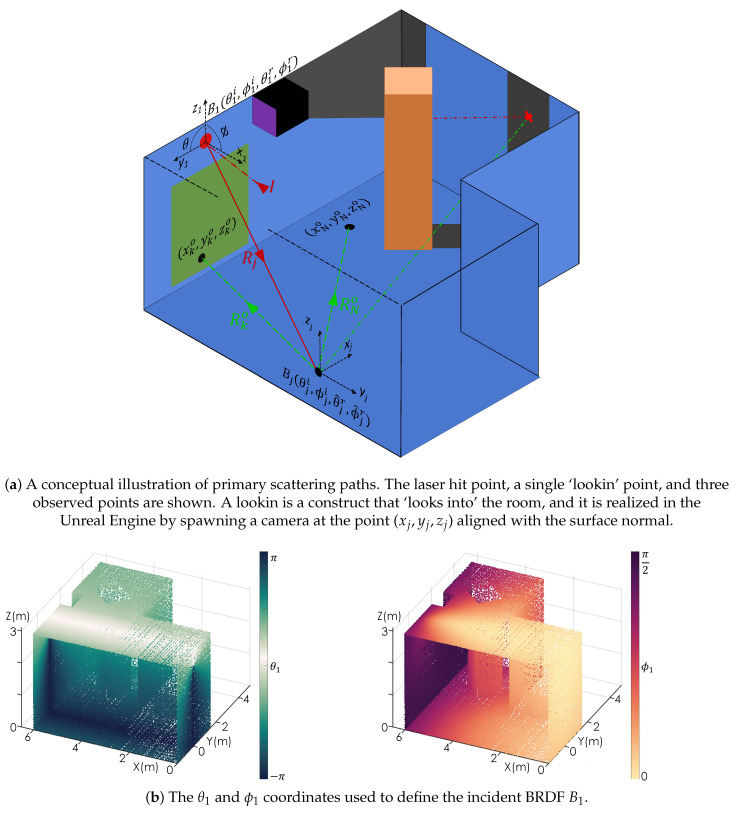
The ray projection scheme from the laser’s incident point to a lookin point: (**a**) The incident laser pulse *I* and one of its scattered rays $R_{j}$ are shown in red. Three scattering paths and observed points are shown in green, originating from a single lookin point. The red cross indicates an observed point, which is in shadow. (**b**) The $\phi_{1}^{r}$ and $\theta_{1}^{r}$ angles, respectively. The $\theta_{1}^{r}$ angle spans a 2π range in the local $y_{1}$, $z_{1}$ plane. The $\phi_{1}^{r}$ angle spans the range [0,π/2] defined relative to the surface normal $x_{1}$.

**Figure 3 sensors-24-06522-f003:**
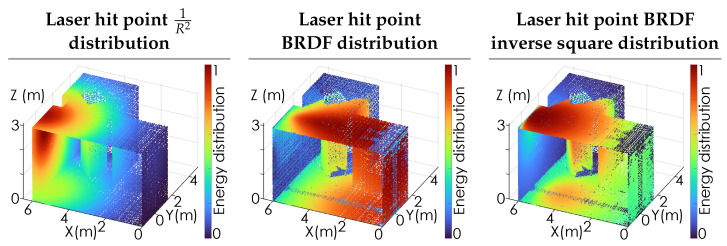
The distribution of the laser energy from the laser hit point throughout the room. The first panel shows the $1/R^{2}$ propagation loss while the second panel shows the Lambertian scattering distribution. The third panel shows the product of the two loss mechanisms, illustrating which points in the room receive the largest fraction of direct laser illumination.

**Figure 4 sensors-24-06522-f004:**
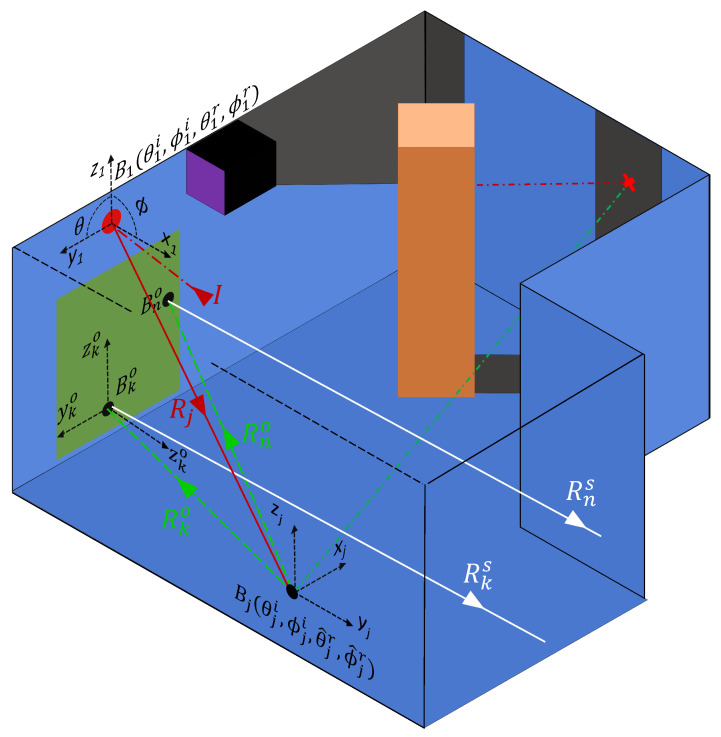
The implementation of the primary scattering paths. For each lookin point ($x_{j},y_{j},z_{j}$) the local incident angles ($\theta_{j}^{i},\phi_{j}^{i}$) from the laser point are calculated. Additionally, the reflection angles (${\hat{\theta }}_{j}^{r},{\hat{\phi }}_{j}^{r}$) to each observed point ($x_{k}^{o},y_{k}^{o},z_{k}^{o}|k\in \left[1,\tilde{N}\right]$) are calculated. The BRDF, abbreviated as “*B*", for each lookin $B_{j}$ is calculated using the local angles. Some fraction of the observed points for each lookin fall within the FoV of the SPAD. The energy from this reduced set of observed points is scattered back to the sensor. This ‘final’ scattering is proportional to the BRDFs and Γ’s of the observed points within the FoV ($\left[B_{k}^{o}:B_{n}^{o}\right]$) as well as the path lengths back to the sensor $R_{k}^{s}|k\in \left[1,n\right]$.

**Figure 5 sensors-24-06522-f005:**
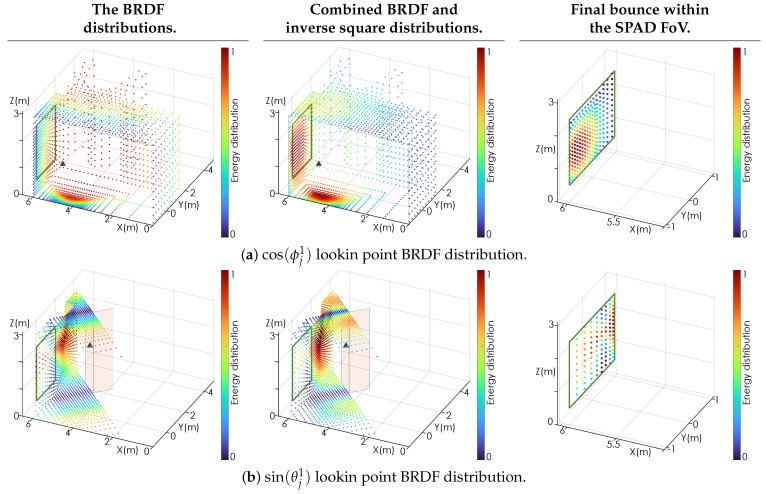
(**a**) The energy distribution for a lookin point on the nearest wall, shown as a gray triangle, with a Lambertian BRDF. (**b**) The energy distribution for a lookin point on the pillar, shown as a gray triangle, with a sin(θ) BRDF. The green square illustrates the FoV of the SPAD. The observation points that fall within the green square are used to calculate the final bounce back to the SPAD. Left column: the BRDF energy distributions from their respective lookin points. Center column: the effect of the inverse square scaling on the energy distributions. Right column: close-in views of the SPAD FoV. These panels also illustrate the relative scattering intensity of each observed point including the final bounce calculation.

**Figure 6 sensors-24-06522-f006:**
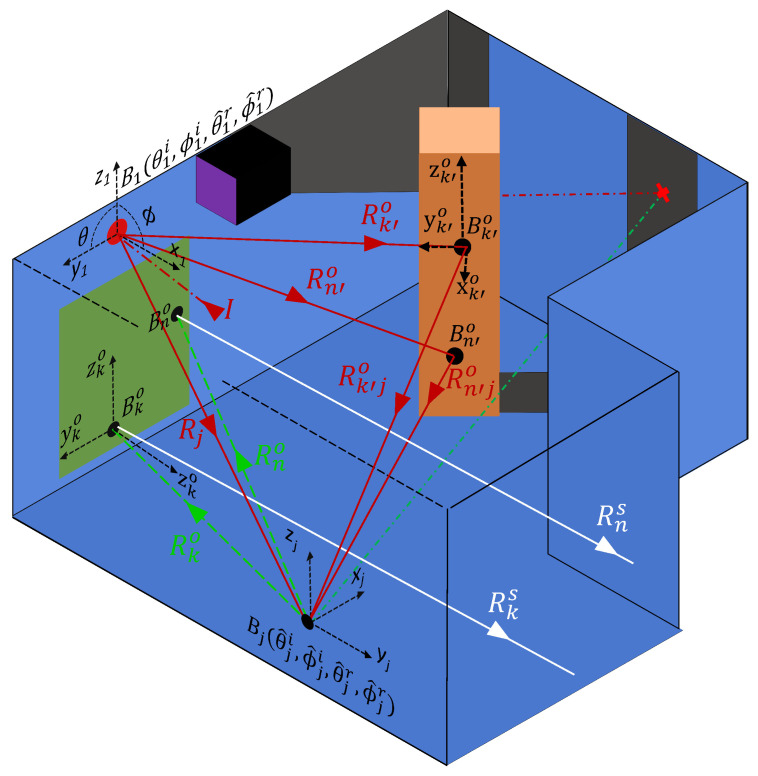
The implementation of secondary scattering paths. For each lookin point ($x_{j},y_{j},z_{j}$) the local incident angles are expanded to a set (${\hat{\theta }}_{j}^{i},{\hat{\phi }}_{j}^{i}$) corresponding to each illuminated (i.e., not in shadow) observed point ($x_{k^{\prime }}^{o},y_{k^{\prime }}^{o},z_{k^{\prime }}^{o}|k^{\prime }\in \left[1,\tilde{N}\right]$). Each of these observed points contributes a scattering component proportional to the BRDFs of the observed points ($B_{k^{\prime }}^{o}|k^{\prime }\in \left[1,\tilde{N}\right]$), their coefficients of reflection $\Gamma_{k^{\prime }}^{o}|k^{\prime }\in \left[1,\tilde{N}\right]$, and the total path length $R_{k^{\prime }}^{o}+R_{k^{\prime }j}^{o}|k^{\prime }\in \left[1,\tilde{N}\right]$ from the laser hit point to the lookin point.

**Figure 7 sensors-24-06522-f007:**
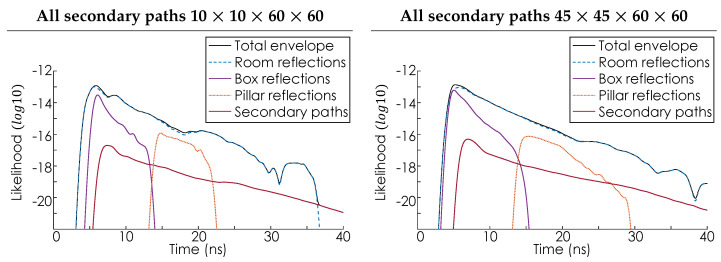
The panels show the likelihood envelope function L(t), plotted on a log scale, for a 10×10×60×60 and a 45×45×60×60 sampling configuration, respectively. The likelihood envelope function L(t) is the uppermost solid black line. The color−coded lines illustrate the contributions to the total envelope of each feature of the room. Specifically, the walls, roof, and floor (blue), the box (purple), the pillar (orange), and the secondary bounce (red).

**Figure 8 sensors-24-06522-f008:**
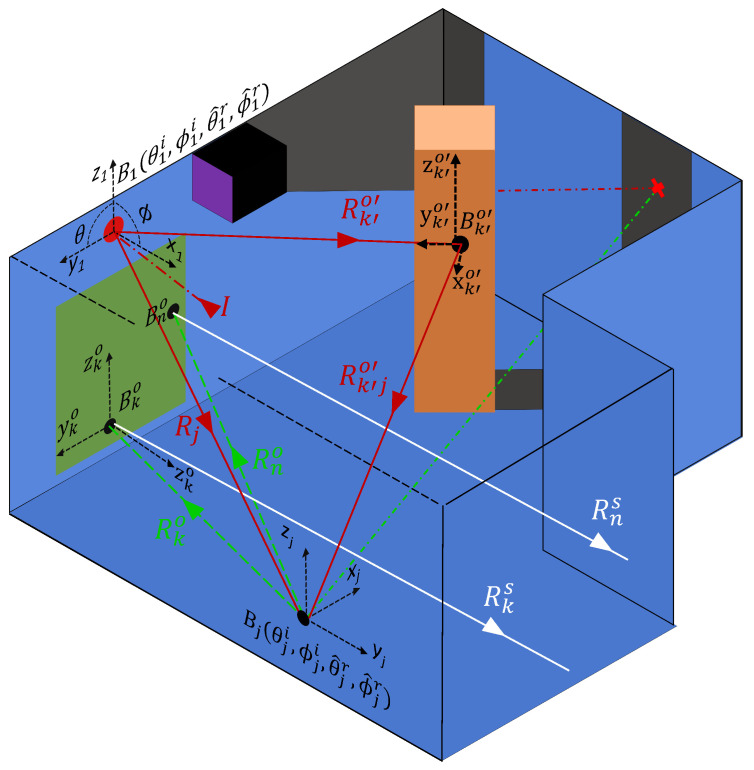
The optimized secondary scattering mechanism. For each lookin point ($x_{j},y_{j},z_{j}$) only the observed point with the largest contribution ($x_{k^{\prime }}^{o^{\prime }},y_{k^{\prime }}^{o^{\prime }},z_{k^{\prime }}^{o^{\prime }}$) is considered.

**Figure 9 sensors-24-06522-f009:**
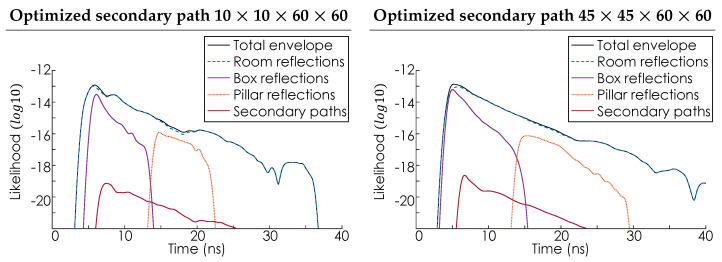
The panels show the likelihood envelope function L(t) for the optimized secondary scattering mechanism, plotted on a log scale, for a 10×10×60×60 and a 45×45×60×60 sampling configuration, respectively. The likelihood envelope function L(t) is the uppermost solid black line. The color−coded lines illustrate the contributions to the total envelope of each feature of the room. Specifically, the walls, roof, and floor (blue), the box (purple), the pillar (orange), and the secondary bounce (red).

## Data Availability

We have made the data used in this work publicly available at https://github.com/HWQuantum/NLOS-SIM, (accessed on 12 September 2024).

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
