# Peer review of "Path Tracing-Inspired Modeling of Non-Line-of-Sight SPAD Data"

_sensors, 2024, doi:10.3390/s24206522_

Round 1

Reviewer 1 Report

Comments and Suggestions for Authors

The manuscript is an extension to the author’s earlier experimental work on non-line-of-sight imaging. The novelty lies in the creation and testing of a modelling approach that enables testing of the expected performance of the method in a range of different scenarios without having to create experimental implementations for each.

The manuscript is well-written and supports repeatability of the results. A lot of the data is difficult to visualize effectively (e.g. Fig. 5) and I am satisfied with the quality of the figures, except for a few formatting issues detailed below.

It is great that the authors achieved a reduction of the computational cost from O(N2) to O(N). Can you describe what this means in real terms? How much quicker did the calculations for Figure 9 take in comparison to those for Figure 7? Will the main advantage be that a larger number of scenarios can be tested in the same time, or will it rather be that each scenario can be tested with a better fidelity?

There are a few small formatting issues relating to the figures:

In figs 7 and 9: “1x10” on the y-axis should probably read “log10”, if I understand correctly. Line styles in legend and figure do not match perfectly (e.g. box reflections is dashed in legend but solid line in figure) and are difficult distinguish unless zoomed in strongly.

Several plots (e.g. figure 5) do not have units on the x/y/z axes (I believe this is in meters).

There is some redundancy between figure captions and subfigure captions.

Overall, I believe this is a useful manuscript and support publication.

Reviewer 2 Report

Comments and Suggestions for Authors

The paper addresses a highly relevant topic in NLOS imaging using SPAD detectors, particularly in fields like autonomous driving and surveillance. By employing a path tracing-inspired simulation framework within the Unreal Engine, the authors present a novel approach to model the temporal distribution of photons in dToF LiDAR-based NLOS imaging. The optimized secondary scattering method proposed significantly reduces computational complexity while preserving critical physical details, showcasing the innovation in the approach.

While the simulation framework is powerful, it may have limited applicability to real-world physical effects beyond virtual environments. The performance or constraints in complex and dynamic scenes are not explicitly discussed. One of the manuscript's major limitations is the lack of real-world experimental validation. While the author claimed that the framework is consistent with prior experimental data. But the specific experimental details and results are not provided. In addition, a quantitative comparison between the simulation and experimental results is also needed

more emphasis should be placed on this limitation and how it could be addressed in future work.
